# OpenReview forum: "Typography Leads Semantic Diversifying: Amplifying Adversarial Transferability across Multimodal Large Language Models"
_ICLR.cc/2025/Conference — ICLR 2025 Conference Withdrawn Submission_

### Official Review · Reviewer_PjGc · 2024-10-19

**Soundness:** 2
**Presentation:** 2
**Contribution:** 2
**Rating:** 3
**Confidence:** 4

**Summary:**

This paper explores the security vulnerabilities of Multimodal Large Language Models (MLLMs), with a focus on the transferability of adversarial examples. The authors introduce the Multi-semantic Angular Deviation Score (MADS), a quantitative metric for analyzing the adversarial transferability of different image samples. Additionally, they propose the Typography Augment Transferability Method (TATM), which enhances adversarial transferability by leveraging information diversity and cross-modal editing. Through experiments, the authors demonstrate the effectiveness of TATM in improving adversarial transferability.

**Strengths:**

The strengths of this paper include:
- The authors' proposal of a white-box attack method to enhance the transferability of adversarial examples. They conducted experiments on various tasks, such as "Harmful Word Insertion" and "Important Information Protection," thoroughly analyzing the performance of different methods across various MLLMs, including those with fixed vision encoders and cross-vision encoders.

- The introduction of the Multi-semantic Angular Deviation Score (MADS) contributes to improving the interpretability of adversarial examples, offering a valuable tool for understanding their behavior.

**Weaknesses:**

The weaknesses of this work include:
- The authors introduce two real-world applications: "Harmful Word Insertion" and "Important Information Protection." However, the paper does not provide a clear explanation of the specific setups for these applications. Both Figure 1 and the accompanying text fail to clarify these aspects. Additionally, the authors do not justify the choice of "Suicide" and "Unknown" as target outputs, leaving their rationale unclear. Moreover, the explanations for the baseline methods, including DIM, BC, SIM, SIA, and TIM, are also insufficiently detailed.

- The proposed adversarial example generation method is essentially a white-box attack, where the image is initialized using typographic words. However, the advantage of this initialization is not demonstrated. The authors neglect to provide ablation studies, such as comparing adversarial training using original image for initialization or image patch for initialization.

- The authors' exploration of the different word types of typographic words embedded in images lacks clear justification. Intuitively, if the target output is "suicide," the typographic words used in the images should be contextually relevant words or related images. Without such relevance, the significance of testing transferability across arbitrary word types becomes questionable.

- The paper also suffers from numerous typos, indicating a need for significant improvement in writing quality. For instance:

1. Line 124: "Furthermore, data-augmentation methods Data augmentation has received more attention because of the ease and efficiency of implementation."
2. Line 195: Multi-semantic Angular Deviation Score (MADS), and Line 206: "we present the Mean Absolute Deviation Scores (MADS)"—the inconsistent use of different full forms for the same acronym is unacceptable.
3. In Algorithm 1, there are multiple issues with inconsistent capitalization, symbol notation, and typographical errors, including "TypoT."

**Questions:**

Refer to Weakness

---

> ### Author Response · Authors · 2024-12-01
>
> Dear Reviewer PjGc:
>
> We greatly appreciate your detailed review of the paper and valuable review comments. These suggestions will help us further improve our paper.
>
> ### **W1.1: Explanation of Harmful Word Insertion (HWI) and Important Information Protection (IIP) application**
>
> **A1.1:**  We highly value your concern regarding this issue and have further elaborated on the potential societal impact of HWI and IIP in Lines 80-87. Intuitively, HWI can be seen as a jailbreak, where harmful words outside the safety boundary are used as fixed outputs, thereby posing risks to users and society.
>
> For the specific task objectives of jailbreak and the current research progress, please refer to Lines 128-132 in Section 2 Related Work. For the IIP task, we are inspired by guardian algorithms [1-3] that effectively protect image privacy and ownership in Image-to-Image tasks. [1-3] protect the original image content by preventing the visual encoder from reading the input information.
>
> ### **W1.2: The reason for selecting "suicide" and "unknown"**
>
> **A1.2:** Regarding the selection of "suicide", the current direct use of MLLMs involves private interactions with individual users. The target word "suicide", as a harmful piece of information to users, has always been a key focus in jailbreak research, including the literature cited in Lines 128-132. In addition, "suicide" has recently become the first AI jailbreak term in the world to directly cause harm to users [4,5]. Therefore, suicide easily becomes the preferred target word for HWI tasks in jailbreak-like scenarios on MLLMs.
>
> About IIP task, "unknown", as the most intuitive semantic term, is chosen to avoid revealing information about the image and became our primary consideration. In the initial tests, we also conducted small-scale experiments with terms like "protect", "unseen", and "secure" as target words, but the results were not as effective as "unknown". Therefore, we ultimately chose Unknown as the primary approach for the IIP task.
>
> Thank you again for your valuable suggestions. We will further refine the description of this section in subsequent versions.
>
> *[1] Safe Latent Diffusion: Mitigating Inappropriate Degeneration in Diffusion Models, CVPR 2024*
>
> *[2] Can Protective Perturbation Safeguard Personal Data from Being Exploited by Stable Diffusion?, CVPR 2024*
>
> *[3] Latent Guard: a Safety Framework for Text-to-image Generation, ECCV 2024*
>
> *[4] https://www.nytimes.com/2024/10/23/technology/characterai-lawsuit-teen-suicide.html*
>
> *[5] https://edition.cnn.com/2024/10/30/tech/teen-suicide-character-ai-lawsuit/index.html*

---

> ### Author Response · Authors · 2024-12-01
>
> ### **W1.3: The detailed explanations for the baseline methods**
>
> **A1.3:** We apologize for any confusion caused. Regarding the different baseline methods, we initially referred to several related articles on data augmentation [1-4] and adopted the descriptive approach used in those studies for the comparison methods. The schematic of different data augmentation methods and their specific augmented objects (left subfigure of Figure 1), the visualization of an input image processed by different augmentation methods (Figures 2, 9-11(a)), and the references for related comparison methods (Lines 427-429) are provided.
>
> Thank you again for your valuable suggestions on this issue. We have made corresponding improvements in the updated PDF version, including adjusting the size, layout, and resolution of Figures 2, 9-11 to enhance the clarity of each subfigure. We have added the relevant references in Lines 191-193 and provided a more detailed description of each method in Section 6.2 in the appendix. We will also keep refining this section in future versions.
>
> ### **W2: Comparing TATM with original image and image patch initialization**
>
> **A2:** Using the original image and image patches as initialization corresponds to the baseline method (base method) and AIP (Add Image Patch) in the paper. In Section 4 and the appendix, all experimental results related to TATM include comparisons with base and AIP.
>
> *[1] Improving the Transferability of Adversarial Examples with Arbitrary Style Transfer, CVPR 2021*
>
> *[2] Improving the Transferability of Adversarial Samples with Adversarial Transformations, CVPR 2021*
>
> *[3]  Set-level Guidance Attack: Boosting Adversarial Transferability of Vision-Language Pre-training Models, ICCV 2023*
>
> *[4] SA-Attack: Improving Adversarial Transferability of Vision-Language Pre-training Models via Self-Augmentation*

---

> ### Author Response · Authors · 2024-12-01
>
> ### **Q3: Evaluation on relevant context**
>
> **A3:** We considered the use of contextually relevant words as the typography in our initial experimental setting process and also conducted some small-scale validation. Regarding "suicide" and "unknown", we identify and obtain the relevant word sets A and B (Set A includes terms like "kill", "hurt", "blood", etc.; Set B includes terms like "unseen", "security", "privacy", etc.).
>
> We first use "suicide" and "unknown" as target words for the transfer attack and employ the corresponding contextually relevant sets A and B as the typographic text in TATM. In this case, using semantically related words as typographic text accelerates the approach toward the target output, improving TATM performance under the relevant target words. However, when Sets A and B are used as typographic text for the semantically irrelevant words "unknown" and "suicide" in TATM, the overall transferability is nearly zero.
>
> The underlying reason for this phenomenon is mentioned repeatedly in lines 21-23, 46-49, and 139-141 of the original text, where Factor 1—***degree of information diversity***—is identified as a key factor influencing transfer performance. During the process of generating adversarial examples with TATM, enhancing the overall input's information diversity by using different target-irrelevant words as typographic text is key to improving transferability. If the same semantically related words are consistently used, they fail to effectively enhance overall information diversity and may lead to TATM overfitting to specific attack target word scenarios, limiting its generalization ability and application scope.
>
> In addition, compared to manually selecting or using tools to generate synonyms for typographic text targeting specific words, randomly choosing label sets from any dataset or real-world vocabulary significantly reduces the operational complexity of TATM. This, in turn, enhances its social harm in HWI and improves the efficiency of important information protection in IIP.
>
> ### **Q4: Writing**
>
> **A4:** Thank you for your careful reading of the paper and for pointing out the issues. We have addressed these fundamental writing problems in the updated version and will continue to improve them in future versions.

---

### Official Review · Reviewer_vmiD · 2024-10-29

**Soundness:** 2
**Presentation:** 1
**Contribution:** 2
**Rating:** 3
**Confidence:** 4

**Summary:**

The paper investigates vulnerabilities in Multimodal Large Language Models to transferable adversarial attacks. The authors introduce the Typography Augment Transferability Method (TATM), which uses typographic augmentation and cross-modal editing to enhance adversarial transferability across models. TATM proved highly effective in tasks like harmful word insertion and information protection.

**Strengths:**

- The transferability of adversarial attacks across MLLMs is a timely and essential area of study, especially as MLLMs are increasingly integrated into commercial products.

- Comprehensive experiments involving a substantial number of models have been conducted, which is critical for advancing research in transferability studies.

**Weaknesses:**

- The paper requires significant proofreading; several errors disrupt readability and make the reading process frustrating.
The Abstract contains grammatical issues, for example: "Therefore, this paper as the first step to ..." should likely read "this paper serves as the ...". Also, "Furthermore, leveraging two key factors that influence transferability performance: 1) The strength of information diversity involved in the adversarial generation process; 2) Editing across vision-language modality information. We propose a boosting method ..." lacks fluency.
In the Introduction, citations, particularly those listed in groups, should be enclosed in brackets for clarity.
In the Background section, adequate spacing is needed between method names and author names, e.g., "Projected Gradient Descent (PGD)Madry et al. (2017)" should read "Projected Gradient Descent (PGD) Madry et al. (2017)."

- Numerous acronyms and newly introduced terms make the paper challenging to follow.

- Transferability studies should ideally test transferability on black-box production models, such as GPT-4, Gemini, and real deployed systems.

- The method’s dependence on typographic augmentation may restrict its applicability to certain scenarios or datasets. Exploring other forms of semantic augmentation could improve its generalizability and broaden potential applications.

**Questions:**

Please refer to the weaknesses.

**Details Of Ethics Concerns:**

The paper does not appear to address ethical considerations, particularly given the focus on transferability, which could have real-world implications for production models.

---

> ### Author Response · Authors · 2024-12-01
>
> Dear Reviewer vmiD:
>
> Thank you for your valuable review comments. These suggestions will be instrumental in further improving our paper.
>
> ### **Q1 & Q2: Writing**
>
> **A1&A2:** Thank you for your corrections regarding the writing details in the paper. We have addressed these issues in the new version. Additionally, we have simplified many abbreviations and new terms. We will continue to improve these two aspects in future versions. Once again, thank you for your valuable suggestions.
>
> ### **Q3: Evaluation on GPT-4, Gemini, and other real deployed systems**
>
> **A3:** In the experimental setting process, we also considered validating the adversarial transferability of commercial closed-source MLLMs such as GPT-4, Gemini, and Claude. Unfortunately, the test results were not as effective as those on the open-source models.  Open-source MLLMs cannot be compared to commercial MLLMs in terms of model architecture, size, the scale of image-language pairs used for training, and training resources.
>
> Therefore, it is anticipated that using open-source MLLMs as the surrogate model to generate adversarial examples and performing transfer attacks on commercial closed-source models would result in poorer performance. Furthermore, in current studies on transfer attacks in MLLMs [1-3], commercial closed-source models have not been established as a recognized standard for validating transferability.
>
> Additionally, at the practical application level, many applications based on open-source MLLMs already exist (e.g., LLaVA). For example, AgroGPT [4] has been applied in agricultural tasks, [5-6] in autonomous driving, and [7-8] in robotic manipulation. Therefore, studying the adversarial transferability between open-source MLLMs is of significant practical importance.
>
> *[1] On Evaluating Adversarial Robustness of Large Vision-Language Models NeurIPS 2023*
>
> *[2] An Image is Worth 1000 Lies: Adversarial Transferability Across Prompts on Vision-Language Models, ICLR 2024*
>
> *[3] Exploring the Transferability of Visual Prompting for Multimodal Large Language Models CVPR 2024*
>
> *[4] AgroGPT: Efficient Agricultural Vision-Language Model with Expert Tuning*
>
> *[5] Feedback-Guided Autonomous Driving, CVPR 2024*
>
> *[6] DriveGPT4: Interpretable End-to-End Autonomous Driving Via Large Language Model, IEEE ROBOTICS AND AUTOMATION LETTERS*
>
> *[7] OpenVLA: An Open-Source Vision-Language-Action Model*
>
> *[8] LLaRA: Supercharging Robot Learning Data for Vision-Language Policy*

---

> ### Author Response · Authors · 2024-12-01
>
> ###  **Q4: Generalizability and broaden potential applications**
>
> **A4:** Thank you for considering the concern of "generality and potential application expansion". TATM is essentially a data augmentation-based method for enhancing adversarial transferability. Data augmentation has gained widespread attention due to its ease of implementation and the convenience of testing across different datasets, models, and scenarios. Regarding the complexity of the TATM implementation, as outlined in Algorithm 1 (Lines 272-290), we enhance the transferability of the final adversary by randomly selecting typographic text and applying basic image processing techniques to print it onto the target image during each iteration of adversarial example generation (Lines 279).
>
> Additionally, the selection factors for typographic text can also be guided by the discussions in Figure 1, Figure 2, and [1].  Therefore, under the corresponding guidance, TATM can enhance adversarial transferability with minimal cost by augmenting the image. Regarding the concern of **restrict its applicability to certain scenarios or datasets**, the test images selected in our experiments prioritize strong representativeness and high generalizability. In Section 4.1 Datasets (Line 316-322),  the test images are randomly selected from various object and material categories in the widely recognized MS-COCO dataset to ensure diversity in the test subjects. In a word, TATM demonstrates high flexibility and generalizability, both in its algorithm implementation and in the test results of representative examples.
>
> *[1] Unveiling Typographic Deceptions: Insights of the Typographic Vulnerability in Large Vision-Language Model, ECCV 2024*

---

### Official Review · Reviewer_zGGc · 2024-11-03

**Soundness:** 3
**Presentation:** 2
**Contribution:** 3
**Rating:** 6
**Confidence:** 4

**Summary:**

This paper investigates the transferability of adversarial examples across MLLMs. Although MLLMs excel in cross-modal interaction and comprehension, they remain vulnerable to transferable adversarial attacks, posing significant real-world risks. Leveraging two key factors—information diversity in adversarial generation and cross-modal editing—the authors propose the Typography Augment Transferability Method (TATM), which enhances adversarial transferability across MLLMs. Experiments demonstrate TATM’s effectiveness in applications such as "Harmful Word Insertion" and "Important Information Protection.

**Strengths:**

1. the authors propose an innovative approach to enhance transferability by leveraging cross-modal data augmentation and semantic diversification through typography-based adversarial examples.
2. The introduction of the Multi-semantic Angular Deviation Score (MADs) as a metric for quantifying information diversity also reflects the technical rigor of the study.

**Weaknesses:**

1. There is a lack of in-depth analysis on the transferability of adversarial examples. For instance, to further understand how the targeted adversarial example influences response generation, the authors can compute the relevancy score of image patches related to the input question using GradCAM to obtain a visual explanation for both clean and adversarial images.
2. Reference missing, such as "On Evaluating Adversarial Robustness of Large Vision-Language Models"

**Questions:**

1. Although the authors have validated the proposed method on multiple models, the experiments are limited to fixed-size models and scenarios. Expanding the diversity of experiments is recommended, such as testing larger models like Qwen2-VL-8B, CogVLM-17B, and Yi-VL-34B, to further assess the method's effectiveness.

2. What is the impact of the perturbation budget on the transferability of adversarial examples in applications like "Harmful Word Insertion" when evaluated on larger models? Generally, larger MLLMs exhibit more robust visual understanding.

3. "On Evaluating Adversarial Robustness of Large Vision-Language Models" is likely the first work investigating the adversarial robustness of MLLMs. Could this method be used as a baseline?

---

> ### Author Response · Authors · 2024-12-01
>
> Dear reviewer zGGc:
>
> We are very grateful for your detailed and highly constructive suggestions. We will try our best to respond to these important comments below.
>
>
> ### **W1: Grad-CAM on various adversarial images**
>
> **A1:** In the updated PDF version, we have added the corresponding Grad-CAM visualizations in subfigures (c-e) of Figure 2 and Figures 8-11. (c) and (d) show the Grad-CAM attention response maps for the adversarial examples generated with "suicide" as the target output, under the original language output and the target output "suicide". (e) shows the Grad-CAM attention response map for the adversarial examples generated with "unknown" as the target output, under the original language output. The target output 'Unknown' is not used for the attention map here, as the Important Information Protection (IIP) task focuses more on protecting privacy by deviating from the original information, rather than requiring the specific target output 'Unknown.'
>
> In addition, in Section 3 Adversary Performance (lines 290-298), we further elaborate on the different behaviors observed in (c)-(e). Compared to the adversarial examples generated directly from clean images (base method), the uni-semantic method primarily focuses on the original subject object (highlights still concentrated on or around the cat). In contrast, the attention scope of the multi-semantic methods is broader, tending to cover the entire visual range (the highlights are more dispersed across the image). Furthermore, compared to Admix and AIP, this trend is more pronounced in TATM (the highlights have the broadest distribution across the entire image). The same behavior is observed in (c)-(e) of Figures 8-11. This further confirms that, compared to Clean and other augmentation methods, TATM introduces a higher level of information diversity by incorporating text semantics into visual information.
>
>
> ### **W2 & Q3: Reference of "On Evaluating Adversarial Robustness of Large Vision-Language Models"**
>
> **A2&A3:** Thank you for the reminder. In fact, we have previously cited [1] in lines 121-122. [1] is a very insightful work that first reveals the cross-model transfer attack vulnerability between different VLMs and further enhances the transferability through black-box querying methods.  As we emphasize in the related work section, this paper can be considered the cornerstone of our current work.
>
> Additionally, the baseline method we primarily compare, **base**, uses the same transfer attack algorithm as [1]. Therefore, [1] should be considered a key comparison method, and we will clarify this in subsequent versions. However, from the perspective of research focus (transferability across MLLMs) and application method (data augmentation-based), [1] kindly differs slightly from the current work. In future updates, we will integrate the highly constructive work from [1] into our evaluation framework. For example:
> ###### Performance comparison of TATM with other data augmentation methods under the surrogate and victim model settings like [1].
> ###### Exploration of whether combining TATM with black-box querying can further enhance the transferability of adversarial attacks.
>
> *[1] On Evaluating Adversarial Robustness of Large Vision-Language Models, NeurIPS 2023*

---

> ### Author Response · Authors · 2024-12-01
>
> ### **Q1 & Q2: More diverse exerpiemnts**
>
> **A1&A2:** Many thanks for your valuable feedback on the effects of model size and perturbation budgets in adversarial transferability. In response to your concern, we evaluate 300 adversarial examples (targeting "suicide" in the scenario of Harmful Word Insertion) generated by LLaVA-v1.5-7B with TATM on larger models like Qwen2-VL-7B, CogVLM-17B, and Yi-VL-34B. However, since these larger models use different vision encoders (which are a vision transformer with 675M parameters, EVA2-CLIP-E, and CLIP ViT-H/14, respectively) compared to LLaVA-v1.5-7B (with CLIP-ViT-Large-Patch14-336), the adversarial examples cannot attack them successfully.
>
> Additionally, we conduct experiments using Qwen2-VL-7B as the surrogate model to generate 300 adversarial examples targeting "suicide" in the Harmful Word Insertion scenario. These adversarial examples are then evaluated on Qwen2-VL-2B and Qwen2-VL-72B. However, since all three models are trained with unfrozen vision encoders, resulting in distinct vision encoders, the adversarial examples generated using Qwen2-VL-7B fail to transfer successfully to either Qwen2-VL-2B or Qwen2-VL-72B.
>
> Therefore, we evaluate LLaVA-v1.6-34B, another larger model that utilizes the same vision encoder as LLaVA-v1.5-7B. The following Attack Success Rate (ASR) results demonstrate that LLaVA-v1.6-34B has the lowest ASR among all models tested, exhibiting more robust visual understanding.
>
> | Method |   LLaVA-v1.6-7B  |   LLaVA-v1.6-13B  |   LLaVA-v1.6-34B  |
> |:------:|:------:|:------:|:------:|
> |  base | 0.006 | 0.006 | 0.006 |
> |  TATM | 0.169 | 0.152 | 0.053 |
>
> We also evaluate the effect of perturbation budgets on adversarial transferability. Specifically, in the Harmful Word Insertion scenario targeting "suicide", we use InstructBLIP-Vicuna-7B as the surrogate model and test TATM with different perturbation budgets: 4, 8, 12, and 16. The following Attack Success Rate (ASR) results demonstrate that larger perturbation budgets bring stronger adversarial transferability.
>
> | Perturbation Budgets | BLIP2-T5-XL | BLIP2-T5-XXL | BLIP2-OPT-2.7B | BLIP2-OPT-6.7B | InstructBLIP-T5-XL | InstructBLIP-Vicuna-13B | Minigpt4-v1-Llama2-7B | Minigpt4-v1-Vicuna-7B |
> |:--------|:------------|:--------------|:---------------|:---------------|:------------------|:---------------------|:--------------------|:--------------------|
> | 4 | 0.040 | 0.063 | 0.090 | 0.083 | 0.040 | 0.057 | 0.017 | 0.020 |
> | 8 | 0.379 | 0.475 | 0.485 | 0.545 | 0.392 | 0.455 | 0.150 | 0.183 |
> | 12 | 0.409 | 0.591 | 0.555 | 0.605 | 0.465 | 0.545 | 0.292 | 0.272 |
> | 16 | 0.429 | 0.545 | 0.535 | 0.641 | 0.578 | 0.661 | 0.256 | 0.269 |

---

### Official Review · Reviewer_knUv · 2024-11-04

**Soundness:** 2
**Presentation:** 3
**Contribution:** 2
**Rating:** 5
**Confidence:** 3

**Summary:**

The paper investigate the threat of cross-MLLMs adversarial transferability. The paper proposes a boosting method, TATM, leveraging the strength of information diversity involved in the adversarial generation process and editing across the modality information.

**Strengths:**

1. The paper provides comprehensive evaluation on various surrogate model architectures, such as BLIP2, InstructBLIP, LLaVA, MiniGPT4.

2. Provides different attack baselines, which makes the evaluation stronger.

3. The paper is easy to understand.

**Weaknesses:**

1. Does the proposed attack method resistant to the defense mechanism? Are there any analysis or the defense baselines evaluation?

2. The selected tasks are generation-based tasks. Will the attack also works on the classification tasks?

**Questions:**

Please check the weakness

---

> ### Author Response · Authors · 2024-12-01
>
> Dear reviewer knUv:
>
> We are very grateful for the Reviewer knUv’s kind reviews. We will answer the constructive comments accordingly.
>
> ### **W1: Performance under the  defense mechanism**
>
> **A1:** The performance of our TATM method under defense mechanisms is discussed in Section 4.5, where we examine two commonly used techniques to counter current black-box attacks: Gaussian Noise and Gaussian Blur [1-4].
>
> **Gaussian noise (GN)**:
>
> $GN  = x + N  \quad \text{s.t.} \quad N(x) \sim \mathcal{N}(\mu, \sigma_{GN}^2)$
>
> where the mean value $\mu$ is 0 and the standard deviation $\sigma$ is 0.005.
>
> **Gaussian Blur (GB)**:
>
> $O(x, y) = \sum_{i=-k}^{k} \sum_{j=-k}^{k} G(i, j) I(x-i, y-j)$
>
> $G(i, j) = \frac{1}{2\pi\sigma_x\sigma_y} e^{-\frac{i^2}{2\sigma_x^2} - \frac{j^2}{2\sigma_y^2}}$
>
> where the kernel size is 3, and $\sigma$ is 0.1.
>
> In terms of experimental results, Table 3 and Table 5 in the appendix present the performance of our TATM method under different cross-MLLM scenarios (e.g., surrogate model: InstructBLIP-7B, victim model: VM1-VM8; surrogate model: LLaVA-v1.5-7B, victim model: VM9-VM13), as well as comparisons against different target outputs ("suicide" and "unknown") and baseline methods (base, DIM, SIM, BC, TIM, SIA, Admix, AIP). The results demonstrate that TATM generates adversarial examples with stronger adversarial transferability.
>
> *[1] Random Noise Defense Against Query-Based Black-Box Attacks, NeurIPS 2021*
>
> *[2] On the Effectiveness of Small Input Noise for Defending Against Query-based Black-Box Attacks, CVPR 2022*
>
> *[3] Beyond Pretrained Features: Noisy Image Modeling Provides Adversarial Defense, NeurIPS 2024*
>
> *[4]  Towards Evaluating Gaussian Blurring in Perceptual Hashing as a Facial Image Filter*

---

> ### Author Response · Authors · 2024-12-01
>
> ### **W2:  Evaluation on classification task**
>
> **A2:** Actually, MLLMs are designed as generative models to complete a range of cross-modal interactive perception Artificial Intelligence Generated Content (AIGC) tasks. Compared to classification tasks using prompts such as "classify the image object" or "give the category of this image," generative tasks that use prompts like "describe the image content" require a more comprehensive and detailed perception of visual information, thereby highlighting the performance advantages of MLLMs.
>
> However, your suggestions are still highly constructive! To thoroughly evaluate TATM's performance, we used the 79 image-classification prompts from [1] to test its performance on classification tasks. (In each experiment, for each image, one prompt is randomly selected from these 79 to assess cross-prompt performance.) For the comparison, we selected the base method, TATM, and the other two methods with the best performance in Tables 1 and 2 ("suicide": Admix, DIM; "unknown": AIP, BC) from the original text generation tasks. The specific experimental results are as follows:
>
> The Attack Successful Rate (ASR) of Victim Model (VM) 1-8 (Surrogate Model: InstructBLIP-7B) and VM 9-13 (Surrogate Model: LLaVA-v1.5-7B)  when adopting "suicide" as target output:
>
> |       |   VM1  |   VM2  |   VM3  |   VM4  |   VM5  |   VM6  |   VM7  |   VM8  |   VM9  |  VM10  |  VM11  |  VM12  |  VM13  |
> |:-----:|:------:|:------:|:------:|:------:|:------:|:------:|:------:|:------:|:------:|:------:|:------:|:------:|:------:|
> |  base | 0.3256 | 0.2658 | 0.1595 | 0.2193 | 0.2060 | 0.2525 | 0.0831 | 0.0897 | 0.0233 | 0.0266 | 0.0066 | 0.0066 | 0.0066 |
> |  DIM  | 0.6146 | 0.5150 | 0.3289 | 0.4086 | 0.4020 | 0.4319 | 0.0930 | 0.1096 | 0.1296 | 0.1462 | 0.0997 | 0.1362 | 0.1163 |
> | Admix | 0.4917 | 0.4585 | 0.3223 | 0.3555 | 0.3721 | 0.3953 | 0.1229 | 0.1030 | 0.2060 | 0.2159 | 0.1561 | 0.2525 | 0.1960 |
> |  TATM | 0.6478 | 0.7176 | 0.4884 | 0.6744 | 0.5116 | 0.6047 | 0.3123 | 0.2824 | 0.2226 | 0.2359 | 0.1130 | 0.1694 | 0.1528 |
>
> The ASR of VM 1-8 (Surrogate Model: InstructBLIP-7B) and VM 9-13 (Surrogate Model: LLaVA-v1.5-7B)  when adopting "unknown" as target output:
>
> |       |   VM1  |   VM2  |   VM3  |   VM4  |   VM5  |   VM6  |   VM7  |   VM8  |   VM9  |  VM10  |  VM11  |  VM12  |  VM13  |
> |:-----:|:------:|:------:|:------:|:------:|:------:|:------:|:------:|:------:|:------:|:------:|:------:|:------:|:------:|
> |  base | 0.0299 | 0.0166 | 0.0100 | 0.0199 | 0.0133 | 0.0365 | 0.0033 | 0.0100 | 0.0664 | 0.0299 | 0.0399 | 0.0532 | 0.0432 |
> |   BC  | 0.0631 | 0.0399 | 0.0133 | 0.0332 | 0.0332 | 0.0631 | 0.0133 | 0.0233 | 0.1096 | 0.1329 | 0.0532 | 0.1096 | 0.0963 |
> |  AIP  | 0.0864 | 0.0565 | 0.0133 | 0.0399 | 0.0465 | 0.0598 | 0.0166 | 0.0199 | 0.0997 | 0.0831 | 0.0498 | 0.1096 | 0.0698 |
> |  TATM | 0.1628 | 0.1429 | 0.0631 | 0.1130 | 0.0864 | 0.1296 | 0.0598 | 0.0299 | 0.1262 | 0.1329 | 0.0731 | 0.0831 | 0.0864 |
>
> The experimental comparison shows that in the HWI task with "suicide" as the target word, TATM still demonstrates strong ASR performance. Since the vocabulary for language outputs in classification tasks is typically smaller than the long passages in generative tasks, the target word "unknown" also shows a performance advantage in terms of ASR (without the need for CLIPScore, the ASR is generally better than that of the base method and other data augmentation methods).

---

### Author Response · Authors · 2024-12-01

We sincerely appreciate the hard work of all reviewers and their valuable suggestions provided for our paper. Based on the precise review comments, we have updated the main PDF. We kindly look forward to reviewers' further feedback. Very grateful to each reviewer once again!

---

### Author Response · Authors · 2024-12-02

We deeply appreciate the reviewers' dedicated efforts!  According to each reviewer's valuable suggestions, we have made significant improvements to the writing details, paragraph formatting, content organization, and image visualizations in the main PDF. We still hope the updated version addresses your concerns to some extent. If there are any further questions, we welcome additional discussions. Thank all reviewers once more!

---

### Note · Authors · 2024-12-14

I have read and agree with the venue's withdrawal policy on behalf of myself and my co-authors.